# The Effect of Mind Subtraction Meditation Intervention on Smartphone Addiction and the Psychological Wellbeing among Adolescents

**DOI:** 10.3390/ijerph17093263

**Published:** 2020-05-07

**Authors:** Eun-Hi Choi, Min Young Chun, Insoo Lee, Yang-Gyeong Yoo, Min-Jae Kim

**Affiliations:** 1Department of Nursing, Eulji University, Daejeon 34824, Korea; choieh@eulji.ac.kr; 2Department of Global Medical Science, Sungshin Women’s University, Seoul 02844, Korea; 3Department of Paramedic Science, Korea National University of Transportation, Chungcheongbuk-do 27909, Korea; islee@ut.ac.kr; 4Department of Nursing, Kunsan National University, Jeollabuk-do 54150, Korea; ygyoo@kunsan.ac.kr; 5Research Institute of Basic Science, Sungshin Women’s University, Seoul 02844, Korea; mingzae@gmail.com

**Keywords:** smartphone addiction, mind subtraction meditation, adolescent

## Abstract

As the smartphone has become an indispensable device in modern lives, consequential psychosocial problems such as smartphone addiction have been getting attention worldwide, especially regarding adolescents. Based on its positive effect on young individuals’ mental health, mind subtraction meditation has been widely applied to many school-based programs in South Korea. This study aims to identify the effects of a school program based on mind subtraction on the smartphone addiction of adolescents. A total of 49 high school sophomores, 24 from the experimental group (mean age = 16), and 25 from the control group (mean age = 16) are included in this case-control study. The experimental group is given the meditation program sessions in the morning, two times a week for 20 min per session, for a total of 12 weeks. The experimental group shows improvements regarding the ‘smartphone addiction’ section (*p* < 0.001), for instant satisfaction (*p* < 0.001) and long-term satisfaction (*p* < 0.001). Concerning the ‘self-control’ section and decreasing stress (*p* < 0.001), problem focusing (*p* < 0.001), and social support navigation (*p* = 0.018), there are improvements in these ‘stress coping strategies’ sections. This study directly shows the positive effect of mind subtraction meditation on smartphone addiction in adolescents and, thus, provides guidance to the future development of smartphone addiction prevention programs for young individuals.

## 1. Introduction

As mobile technology has been growing rapidly in the last two decades, smartphones have evolved into indispensable devices in our lives. The worldwide population of smartphone users has already exceeded 3 billion in 2019 and is anticipated to grow further [1]. Regarding South Korea, the number of smartphone users has constantly increased from 40.6 million in 2014 to 43.7 million in 2015 [2]. The smartphone dissemination rate among South Korean young individuals also has been rapidly growing, as it is shown to be 81.5% in 2013, which is an increase from 21.4% in 2011 [3].

### 1.1. Smartphone Addiction

Despite the enormous convenience that smartphones have brought into the modern society, they also have bred some serious problems, including smartphone addiction. Overusing the smartphone is thought to be the key behavior of developing an addiction. It affects daily lives significantly via lowering self-control and, eventually, might lead to anxiety, depression, and nervousness [4]. Considering a recent survey, 89.5% of the South Korean population owned a smartphone in 2018 and, among them, 2.7% had a high risk and 15.9% had a potential risk of smartphone addiction [5]. This has been steadily rising since 2011, regardless of the age group, while the adolescent age group had the highest risk percentage of 30.3 [5].

Statistics like these automatically bring much attention worldwide to figure out how smartphone addiction affects young individuals, and several studies have found that it is positively related to psychosocial disabilities. According to a study held in 2017, there were positive relationships between problematic usage of the smartphone and psychopathologic disorders, which appeared the strongest in depression, followed by chronic stress and anxiety [6]. There is also a study stating that Japanese young adults with smartphone addiction tend to suffer from social isolation (i.e., Hikikomori), which is mainly derived from gaming disorders in males and social networking service addiction in females [7].

### 1.2. Mind Subtraction Meditation

Seeking effective prevention strategies for smartphone addiction has been considered an urgent national issue due to its negative effects on adolescents, such as school maladjustment [8,9] or in some psychosocial disorders including depression and anxiety [6]. There have been numerous studies on finding ways to prevent smartphone addiction [10]. Among those which remain effective, mind subtraction meditation is the one which has been widely applied as a school-based program in South Korea [11].

Mind subtraction meditation was founded by Woo and it is based on the holistic concept of focusing on the true nature of human beings and the acceptance of the original mind of the universe (universe mind) [12]. All human beings are born with this universe mind, but memories of personal experiences with emotions, feelings, senses, and perceptions accumulate in their minds [12]. This accumulated mind is a false mind which filters the world through their own point of view, causing one to have misperceptions and misinterpretations about the world outside, leading to stress, anxiety, and depression [12]. Mind subtraction meditation provides a systemic and scientific method, focusing on subtracting the false mind so the original universe mind can emerge [13]. This method is very simple to anyone who continues practicing, and is offered in various settings, such as businesses, government agencies, colleges and youth camps.

Furthermore, mind subtraction meditation was introduced into the schools’ character education program in South Korea by showing students the positive changes in their self-control by reducing anxiety and stress in their inner self through self-reflection [14]. Prior studies using this meditation program in schools also reported significant improvements in depression for youth [15], aggregation and anger for elementary students, and self-control for middle school students [16].

During this study, high school students are offered the mind subtraction meditation program to examine its effect on smartphone addiction, stress levels, self-control, and stress coping strategies. Even though many studies are available supporting the positive effects of meditation, there is no research related to smartphone addiction in adolescents.

### 1.3. Purpose

This present study aims to find the effectiveness of mind subtraction meditation in preventing smartphone addiction, via analyzing the effects of a mind subtraction meditation-based program held in a high school in South Korea, to eventually contribute to the development of a smartphone addiction prevention program for adolescents.

## 2. Materials and Methods

### 2.1. Participants in the Research

This research was based on the self-report questionnaire administered in the school-based meditation program for smartphone users. This study used a nonequivalent group comparison with a pretest and post-test design to examine the effects of a school-based meditation program on smartphone addiction, self-control, stress levels, and stress coping strategies. Due to high academic pressure and competition, most of the high school students from South Korea were unable to participate in after-school programs. Therefore, we had to use convenience sampling which was already assigned to a chosen classroom for the meditation program, rather than designing a randomized study.

This program was performed in the morning self-study time of one high school in a large urban area of South Korea in April 2015. The experimental group was given meditation program sessions by their instructor two times a week for 20 min per session, for a total of 12 weeks. The control group did not receive any intervention, but the students of the control group were allowed to read any books they wanted without using smartphones. Measurements were repeated 4 weeks after completion of the meditation program to examine the sustained effect of the meditation.

A total sample size of 52 subjects was necessary to ensure statistically valid results with a power of 0.80 and a statistical significance level of 0.50. A total of 26 subjects in each of the two groups (experimental group: control group = 1:1) would provide an *F*-statistic power of 0.80 at a *p*-value of 0.05 in a two-group analysis of variance [17]. Occurring at the beginning of the meditation sessions, the participants available for recruitment in this study were 52 high school students in a city in South Korea. 

The participants of the experimental group were 26 high school sophomores in the same class (mean age 16 years), who consented to the research study. The control group subjects were 26 sophomore students in the same class year (mean age 16 years) at the same high school. To minimize the interaction between the two groups, two different classes located far from each other were chosen for this study. 

Parental consent was obtained for all students. However, two members of the experimental group and one of the control group were dropped from this study. Thus, a total of 49 students’ data were included in the final analysis. The 24 students in the experimental group comprised 4 boys and 20 girls, while the 25 students in the control group comprised 4 boys and 21 girls. 

### 2.2. School-Based Meditation

The mind subtraction meditation provided to the experimental group included both lectures and meditation. The experimental group participated in a group program in their classrooms twice a week for 20 min each time during their morning meetings, for a total of 12 weeks beginning in April 2015. The program instructor was a current high school teacher with a license in mind subtraction meditation. The instructor was given information on the needs of an educational intervention against the dangers of smartphone addiction and agreed to participate after hearing an explanation of the purpose and methodology of the study from the researcher.

The intervention program began with an explanation of the principles of the mind, followed by meditation to eliminate negative thoughts. The students eliminated their minds of family, worries about school life, anxiety, anger, and stress. During this process, they were asked to review their feelings when they used smartphones, and to get rid of those feelings (Table 1).

### 2.3. Ethical Considerations

The Institutional Review Board of the School of Transportation at a university in South Korea approved the study (Approval Numbers: KNUT IRB-11). The investigators explained to each subject the study’s purpose, procedures, voluntary participation, confidentiality, the potential risks and benefits, as well as the participant’s right to withdraw from this study without penalty. Only with their full consent was the research study initiated with the high school students.

### 2.4. Evaluative Tools

Questionnaires on smartphone addiction, self-control, stress levels, and stress coping strategies were completed in this study.

#### 2.4.1. Smartphone Addiction Self-Report Scale

To measure the severity of participants’ smartphone addiction, we performed the Korean Smartphone Addiction Proneness Scale [17]. The scale comprises 15 items, with 5 items related to daily life disturbances, 2 items related to directivity to the virtual world, 4 items related to withdrawal, and 4 items related to tolerance. The score ranges from 4 to 45, with each item scored on a three-point scale.

The subfactors included: “Daily life disturbance,” causing problems in their everyday lives due to excessive use of smartphones; “Directivity to virtual world,” where they felt more comfortable in the digital world than in reality; “Tolerance,” which refers to feeling satisfied only when using smartphones more than in the past; and “Withdrawal,” where they felt anxious and nervous when they stopped or decreased the use of smartphones, or even displayed compulsive behavior. Each item was scored on a 4-point Likert scale ranging from ‘Strongly disagree’ to ‘Strongly agree’, where higher scores indicated a greater addiction to smartphones. The total score for all 15 items classified the students into a high-risk group, potential risk group, and general users.

According to a study by the National Information Society Agency (2011) [17], the reliability values of the items on the smartphone addiction scale are as follows: 0.88 for smartphone addiction as a whole, 0.83 for daily life disturbances, 0.68 for directivity to virtual world, 0.77 for withdrawal, and 0.78 for tolerance. The Cronbach’s α values of this study were 0.88 for smartphone addiction as a whole, 0.76 for daily life disturbances, 0.56 for directivity to virtual world, 0.73 for withdrawal, and 0.3 for tolerance.

#### 2.4.2. Self-Control Scale

To assess self-control, the study used a scale revised to fit adolescents translated from the self-control scale by Gottfredson and Hirschi [18] and the Self-Control Rating Scale (SCRS) by Kendall and Wilcox [19]. The scale comprises 20 items, with 10 items concerning long-term satisfaction and 10 items regarding instant satisfaction. The score ranges from 20 to 100, with each item scored on a five-point scale.

Long-term satisfaction assesses whether the subject has good concentration, thinks before acting, controls desires, and can efficiently solve problems. Conversely, instant satisfaction assesses whether the subject is impulsive, has self-centered thinking, and acts before speaking. A higher sum of item scores shows a greater sense of self, indicating a better ability to avoid and endure problematic behavior that gives instant satisfaction or is based on temporary impulse. The Cronbach’s alpha in prior studies was 0.78 [20], and the Cronbach’s alpha in this study was 0.75. 

#### 2.4.3. Stress Coping Strategies

To assess stress coping strategies, the study used the scale developed by Lazarus and Folkman (1984) [21] which was then revised and edited by Kim (1988) through a factor analysis [22], and then revised and edited again by Jeong (2004) [23]. The scale consists of 20 items subdivided into four subcategories of five items each: problem focusing, emotion focusing, social support navigation, and wishful thinking.

Problem focusing refers to coming face-to-face with personal and environmental issues that potentially cause stress and changing them to attack the source of the stress. Emotion focusing refers to controlling the emotional state related to stress or caused by stress. Wishful thinking refers to reacting toward stress by thinking or imagining an ideal situation while physically and temporally distancing oneself from the stress or the situations of stress. Finally, social support navigation refers to asking for help from another person to solve the stressful situation or event. Each item is scored on a four-point Likert Scale from 1 with ‘Strongly disagree’ to 4 with ‘Strongly agree’, where higher scores mean that the subjects commonly use a coping strategy. Cronbach’s α of the stress coping strategies scale from Jeong’s study (2004) [23] was 0.87–0.93, and in this study was 0.87. 

#### 2.4.4. Stress Scale

To assess stress, the study used Visual Analogue (VA), used by Crosby (1988) [24], and then, to assess emotional stress levels, used the edited version by Kang et al. (2012) [25]. The participants were asked to self-assess their current stress levels on a scale of 1–9 on a straight line of 10 cm in length where 0 is ‘no stress’ and 10 is ‘extremely stressed’. Higher scores thus showed greater stress.

### 2.5. Statistical Analysis

We used the statistical package for the social science (SPSS) for Windows version 25.0 (IBM, Armonk, NY, USA) and two-sided *p*-values < 0.05 were considered statistically significant. Regarding homogeneity testing between the two groups, a chi-square test was used to compare gender, religion, and experience of smartphone addiction education. Furthermore, an independent *t*-test was used to compare the quantitative variables between the two groups. 

A non-parametric test was used, as there were fewer than 30 samples in each of the experimental and control groups, and some variables did not satisfy the normality test. Additionally, to analyze the effect of the school-based meditation program, the Wilcoxon signed rank test was used to compare the pretest and post-test test scores of the same group for smartphone addiction, self-control, stress coping strategies, and stress level. Furthermore, Friedman tests were used to examine changes of scores across all three assessments. 

## 3. Results

### 3.1. Homogeneity Testing

Sociodemographic data were used to analyze the homogeneity of the participants. Sex, religion, and smartphone addiction education were homogeneous in both groups; however, there were more students using a mobile phone at school in the experimental group than the control group (*p* = 0.003) (Table 2).

Besides individual uniqueness and dependent variables (tolerance in smartphone addiction, self-reporting scale, stress score), all other variables were homogeneous in both groups. The tolerance score for the smartphone addiction self-report scale (*p* = 0.002) and stress score were higher in the experimental group than the control group (*p* < 0.001) (Table 3).

### 3.2. Differences between before and after the School-Based Meditation in Smartphone Addiction, Self-Control, Stress Coping Strategies, and Stress Scores

The differences in the means of smartphone addiction, self-control, stress, and stress coping mechanisms were analyzed before and after intervention, and between the control group (Cont.) and experimental group (Exp.) (Table 4). 

The total score for smartphone addiction of the experimental group decreased from 33.88 to 29.63 (Z = −3.187, *p* = 0.001). By sub-category, there were decreases in daily life disturbance (*Z* = −2.974, *p* = 0.003), withdrawal (*Z* = −2.275, *p* = 0.023), and tolerance (*Z* = −2.604, *p* = 0.009). The control group did not show any differences before or after the intervention. 

Regarding terms of self-control, the instant satisfaction of the experimental group increased from 34.42 to 37.75 (*Z* = −3.069, *p* = 0.002), and that of the long-term satisfaction in the experimental group from 30.88 to 33.67 (*Z* = −2.891, *p* = 0.004). The control group did not show any differences before or after the intervention.

Concerning terms of stress coping strategies, the problem focusing coping strategy in the experimental group increased from 10.25 to 12.92 (*Z* = −3.271, *p* = 0.001), and the social support navigation coping strategy of the experimental group increased from 12.29 to 14.00 (*Z* = −2.672, *p* = 0.008). There were no differences in wishful thinking and emotion focusing, and no changes before and after the intervention from the control group.

The stress levels of the experimental group decreased from 7.79 to 5.67 (*Z* = −3.711, *p* < 0.001), whereas the control group did not show any change from the initial level of 5.72.

### 3.3. The Long-Term Effect of School-Based Meditation on Smartphone Addiction, Self-Control, Stress Coping Strategies, and Stress Scores

The long-term effects of smartphone addiction, self-control, stress, and stress coping strategies of the experimental and control groups were analyzed (Table 5).

The total scores of smartphone addiction decreased long-term for the experimental group (χ^2^ = 27.482, *p <* 0.001). Taken by subcategory, there were decreases in daily life disturbance (χ^2^ = 25.039, *p <* 0.001), withdrawal (χ^2^ = 7.303, *p* = 0.026), and tolerance (χ^2^ = 18.759, *p <* 0.009). The control group did not show differences before and after the program. 

Regarding terms of self−control, the experimental group showed long-term effects in instant satisfaction (χ^2^ = 33.250, *p <* 0.001) and long-term satisfaction (χ^2^ = 11.109, *p* < 0.001). The control group did not show differences before or after the program.

Considering terms of stress coping strategies, the experimental group showed long-term effects in problem focusing (χ^2^ = 15.258, *p <* 0.001) and social support navigation (χ^2^ = 8.023, *p* = 0.018). The control group showed differences in emotion focusing (χ^2^ = 12.023, *p* = 0.002).

The experimental group showed long-term effects in stress (χ^2^ = 32.519, *p* < 0.001), whereas the control group did not show differences before or after the program.

## 4. Discussion

### 4.1. Influence on Smartphone Addition by Mind Subtraction Meditation 

Recently, school-based meditation programs could be an innovative strategy for students with academic, psychosocial, and behavioral demands. This study is one of the few intervention studies that verifies the effect of a school-based meditation program on the smartphone addiction, stress level, self-control, and stress coping strategies of high school students. The meditation program was offered to the students twice a week for 20 min each time during their morning meetings, for a total of 12 weeks. During this study, we found that mind subtraction meditation led to a significant improvement in the effect of daily life disturbances, withdrawal symptoms, and tolerance caused by smartphone addiction. Even after one month of completing the program, the total scores of smartphone addiction tests were decreased in the experimental group. This beneficial effect of the meditation program also was noted in other research studies related to meditation [11,26,27]. A prior study showed that a school-based mind subtraction meditation had a positive effect on smartphone addiction tendency and the addiction-associated mental health of children in South Korea [11]. 

Mind subtraction meditation is based on the fact that it is possible to face the true self and improve one’s understanding of others by being reminded of one’s memories of life, and then eliminating them [14]. After thinking of stressful situations related to family, school life, friends, and tests, participants then think of their behavior during times of stress (using the internet or smartphones, playing games, etc.) and eliminate these behaviors from themselves. This causes them to get rid of their obsession toward smartphones, changing their attitudes toward smartphones naturally [28]. Meditation is a process of realizing the true self and accepting it to integrate the ego [26], and mind subtraction meditation seeks the ego by getting rid of existing mindfulness. This also is effective for smartphone addiction. Therefore, to improve the effectiveness of interventions against smartphone addiction in adolescents, it is necessary to reinforce their self-awareness and motivations for change and improve their senses of achievement and self-esteem [29]. The results of this study are that the experimental group saw a decrease in stress levels and changes in self-control and stress coping strategies. 

The meditation program of this study, however, did not show a significant change in directivity to the virtual world. Another group program for smartphone addiction did not show significant differences in directivity to the virtual world either, and mentioned the need for future studies of the important factors to identify individual effects [30].

### 4.2. Intervention Strategies for Smartphone Addiction for Adolescents

The physiological reasons for smartphone addiction are similar to those for other types of addiction, such as a decreases in dopamine D2 receptors, the effectiveness of the receptors, or ERP (event-related Potential) brain wave responses [31]. A common example is a successful case report of prescribing a selective serotonin reuptake inhibitor (SSRI)-antipsychotic combination for a subject with high addiction levels due to frequent mutations related to serotonin transporter genes [32]. 

Smartphone addiction also is considered a type of behavioral addiction; it is treated with cognitive-behavioral therapy to reduce the behavior rather than using drug treatment [33]. Taking this perspective, the interventions include identifying the behavior intentions for using the internet or smartphones, time management, improving interpersonal relationships, and participating in alternative activities [31]. Young’s cognitive behavioral therapy (CBT) has proven effective in internet addiction using these factors after 12 sessions [34]. Meta-analysis studies of CBT, however, have reported its effectiveness [35] as well as its ineffectiveness [36]. Concerning Korea, schools ban the use of smartphones due to the excessive use of smartphones by adolescents, and a time management program also has been developed; however, its effectiveness has not yet been proven [28]. There may be various reasons for this. As smartphones are characterized by such traits as ubiquity (freedom of location), immediacy (free from delay), individuality (used and carried by a single person), and multimedia abilities, they are liable to cause addiction. Thus, there are limitations when simply controlling the behavior itself [31].

Intervention strategies for smartphone addiction for adolescents should not only ban the behavior, but also consider the causes. Reasons for the smartphone addiction of adolescents include feelings of despair regarding dreams and hopes, stress from exams, and difficulties in peer relationships. Accompanying the increases in smartphone behavior, adolescents become more depressed and lethargic, experience changes in their biorhythms, and seek virtual relationships [29]. Therefore, interventions for adolescents should consider these psychological factors related to their developmental stage to be effective in controlling smartphone addiction. Effective intervention programs for smartphone addiction that use art, music, and physical activities [30,37,38] have shown consistent effects in allowing the participants to understand themselves, accept others, and become confident in social relationships [39,40].

### 4.3. Influence on Self-Control by Mind Subtraction Meditation

Smartphone interventions for adolescents using group counseling have considered self-control, alternative activities, and maintaining impulse mechanisms to control smartphone addiction and have proven their effectiveness accordingly [41]. Stress and self-control affect the motivations to use smartphones and smartphone addiction [42,43]. Self-control refers to the ability to choose delays to obtain relatively greater values over instant gratification [44]. Accordingly, it is the ability to practice socially acceptable behavior to appropriately start or stop a certain behavior according to the surrounding environment. Self-control in smartphone addiction refers to continuously using smartphones without realizing the passage of time and the difficulty in stopping use midway [45]. Self-control is classified into instant satisfaction and long-term satisfaction. Instant satisfaction involves compulsive behavior and self-centered thinking, whereas long-term satisfaction involves concentration, thinking before acting, and controlling desires [19]. The results of this study showed a decrease in instant satisfaction and an increase in long-term satisfaction. This can be interpreted as indicating that adolescents who have looked back on themselves through meditation objectively reviewed their behaviors in life, increased their self-understanding, and improved their sense of control by reflecting on their own behavior [46,47]. Instant satisfaction also decreased in the control group, which may be a limitation of this study of not controlling for school or domestic education.

### 4.4. Influence on Stress Coping Strategies by Mind Subtraction Meditation

Finally, this study demonstrated the effectiveness of mind subtraction meditation (MSM) in problem focusing and social support navigation among the stress coping mechanisms. Problem focusing means the efforts to tackle the source of stress by facing the stressful problem, and social support navigation are ways to seek help from someone else to resolve stress [22]. This finding supports the results of prior studies as well [1,2,47]. Considering prior studies, stress coping was an important factor in drug addiction treatment, as well as addiction varying depending on stress management [48,49]. However, wishful thinking and emotion focusing did not show a significant effectiveness for MSM, perhaps because mind subtraction eliminates the emotions and imagination of the person in action, thus is not effective in managing imagination emotions and wishes regarding stress. 

### 4.5. Limitations and Future Directions

Several limitations of this study and recommendation for future studies were discussed. First, the size of the sample was small and there was no randomization of groups, which would possibly impact the generalizability of the findings. It is suggested that a larger sample size and randomization of the groups should be considered in future studies. Second, the sample only consisted of Korean high school sophomores, predominantly involving the female gender. This also would affect the generalizability of the findings similar to the first limitation thus, in the future, it is recommended to involve more variety in the student groups, including diverse countries. Third, we were unable to fully control the interaction between the two groups due to possible unexpected relationships between the students. Fourth, this study used only self-reported questionnaires for evaluation in the pretests and post-tests, which may lead to response bias. Follow-up assessments through neurophysiology or pathophysiologic analysis is suggested for future studies. Finally, other psychological factors, such as personality characteristics, academic stress, and various medical conditions, were not documented in this study.

Despite these limitations, this study has significance by it verifying the possibilities of using meditation in community settings, such as public schools, and providing its potential to become a direct intervention to manage smartphone addiction, as well as reinforcing self-control and stress-management capacity.

## 5. Conclusions

This study demonstrated beneficial effects of school-based mind subtraction meditation on high school students, regarding smartphone addiction, stress levels, self-control, and stress coping strategies. The mind subtraction meditation program was effective especially regarding the ability for self-control and stress management, by recognizing and eliminating negative and obsessive emotions through self-reflection. Due to the continuity of these positive effects, we believe that it would be imperative to offer such meditation programs continuously to adolescents in educational and healthcare settings. Also, developing practical strategies or intervention methods based on various meditation programs would be beneficial to communities, via helping adolescents cope with stress, manage self-control, and deal with smartphone addiction. 

## Figures and Tables

**Table 1 ijerph-17-03263-t001:** Schedule of the school-based mind subtraction meditation program.

12 Weeks: Tuesday, Thursday (20 min/session)	Topic	Contents of Meditation Activity
1	Orientation; knowing the mind	Orientation to the program (program purpose and methods)
		Knowing the false and true mind
		Knowing the reasons for subtracting the mind
		Knowing the method of subtraction and to practice
2	Throwing away of thoughts/ misperceptions about family	Talking about events with family
		Finding memories and writing about family
		Subtracting thoughts/misperceptions about family
		Verbalizing feelings after subtraction
3	Throwing away of thoughts/misperceptions about school	Talking about events in school (teachers and peers)
		Finding memories and writing about school life
		Subtracting thoughts/misperceptions about school life
		Verbalizing feelings after the subtraction
4	Throwing away of thoughts of inadequacy and dislikes	Talk about memories of inadequacy and dislikes
		Finding memories and writing about inadequacy and dislikes
		Subtracting thoughts/misperceptions about inadequacy and dislikes
		Verbalizing feelings after the subtraction
5	Throwing away of thoughts of anxiety and worries	Talking about memories of anxiety and worries
		Finding memories and writing about anxiety and worries
		Subtracting thoughts/misperceptions about anxiety and worries
		Subtracting the mind that used to be a smartphone
		Verbalizing feelings after the subtraction
6	Throwing away of anger, irritation, and stress	Talking about memories of anger, irritation, and stress
		Finding memories and writing about anger, irritation, and stress
		Subtracting thoughts/misperceptions about anger, irritation, and stress
		Subtracting the mind that used to be a smartphone
		Verbalizing feelings after the subtraction
7	Throwing away of scary thoughts and fear	Talking about memories of scary thoughts and fear
		Finding memories and writing about scary thoughts and fear
		Subtracting thoughts/misperceptions about scary thoughts and fear
		Subtracting the mind that used to be a smartphone
		Verbalizing feelings after the subtraction
8–12	Throwing away of self and my life	Talking about memories of self and my life
		Finding memories and writing about self and my life
		Subtracting the mind that used to be a smartphone
		Subtracting thoughts/misperceptions about self and my life
		Verbalize feelings after the subtraction

**Table 2 ijerph-17-03263-t002:** Homogeneity test of the sociodemographic characteristics between experimental and control groups.

Characteristics	Categories	Experimental Group (*n* = 24)	Control Group (*n* = 25)	χ^2^	*p*-Value
Gender	Male	4 (16.7)	4 (16.0)	0.004	0.950
	Female	20 (83.3)	21 (84.0)		
Religion	Yes	8 (33.3)	5 (20.0)	1.117	0.291
	No	16 (66.7)	20 (80.0)		
Smartphone addiction education	Yes	0 (0.0)	3 (12.0)	3.068 *	0.080
	No	24 (100.0)	22 (88.0)		

Data are shown as number (%). * Fisher’s exact test.

**Table 3 ijerph-17-03263-t003:** Homogeneity test of variables between experimental and control groups.

Characteristics	Categories	Experimental Group (*n* = 24)	Control Group (*n* = 25)	*t*	*p*-Value
Smartphone addiction	Daily life disturbance	11.96 ± 2.87	10.60 ± 3.08	1.596	0.117
	Directivity to virtual world	3.46 ± 1.06	3.44 ± 0.92	0.065	0.949
	Withdrawal	7.92 ± 2.96	8.04 ± 2.48	−0.158	0.875
	Tolerance	10.54 ± 2.21	8.64 ± 1.89	3.244	0.002
	Total score	33.88 ± 7.19	39.00 ± 7.13	1.543	0.130
Self-control	Instant satisfaction	34.42 ± 4.58	35.96 ± 5.45	−0.756	0.453
	Long-term satisfaction	30.88 ± 5.74	32.08 ± 4.46	−0.822	0.415
Stress coping strategies	Problem focusing	10.25 ± 2.29	11.56 ± 2.60	−1.869	0.068
	Social support navigation	12.29 ± 2.16	12.52 ± 3.31	−0.285	0.777
	Wishful thinking	12.79 ± 1.96	12.20 ± 2.94	0.832	0.410
	Emotion focusing	10.38 ± 2.83	10.24 ± 2.30	0.184	0.855
Stress level		7.79 ± 1.59	5.72 ± 2.19	3.779	<0.001

Data are presented as mean ± standard deviation.

**Table 4 ijerph-17-03263-t004:** The effect of school-based meditation on smartphone addiction, self-control and stress coping strategies and stress score.

Characteristics	Categories		Pre-Test *	Post-Test *	*Z*	*p*
Smartphone addiction	Daily life disturbance	Exp.	11.96 ± 2.87	10.54 ± 2.03	−2.974	0.003
		Con.	10.60 ± 3.08	9.96 ± 2.89	−1.344	0.179
	Directivity to virtual world	Exp.	3.46 ± 1.06	3.13 ± 1.03	−1.710	0.087
		Con.	3.44 ± 0.92	3.36 ± 0.95	−0.535	0.593
	Withdrawal	Exp.	7.92 ± 2.96	6.83 ± 2.32	−2.275	0.023
		Con.	8.04 ± 2.47	7.84 ± 2.29	−0.970	0.332
	Tolerance	Exp.	10.54 ± 2.21	9.13 ± 1.83	−2.604	0.009
		Con.	8.64 ± 1.89	8.44 ± 2.10	−0.690	0.490
	Total score	Exp.	33.88 ± 7.19	29.63 ± 5.72	−3.187	0.001
		Con.	30.72 ± 7.13	29.60 ± 7.25	−1.619	0.105
Self-control	Instant satisfaction	Exp.	34.42 ± 4.58	37.75 ± 5.28	−3.069	0.002
		Con.	35.96 ± 5.45	34.84 ± 6.73	−0.631	0.528
	Long-term satisfaction	Exp.	30.88 ± 5.74	33.67 ± 4.57	−2.891	0.004
		Con.	32.08 ± 4.46	32.92 ± 5.29	−1.126	0.260
Stress coping strategies	Problem focusing	Exp.	10.25 ± 2.29	12.92 ± 3.32	−3.271	0.001
		Con.	11.56 ± 2.60	12.12 ± 2.88	−1.379	0.168
	Social support navigation	Exp.	12.29 ± 2.16	14.00 ± 3.56	−2.672	0.008
		Con.	12.52 ± 3.31	12.68 ± 3.44	−0.421	0.674
	Wishful thinking	Exp.	12.79 ± 1.96	13.25 ± 2.31	−1.027	0.304
		Con.	12.20 ± 2.94	12.00 ± 2.90	−0.396	0.692
	Emotion focusing	Exp.	10.38 ± 2.83	11.67 ± 2.60	−1.880	0.060
		Con.	10.24 ± 2.30	10.80 ± 2.36	−1.340	0.180
Stress level	Stress level	Exp.	7.79 ± 1.59	5.67 ± 2.18	−3.711	<0.001
		Con.	5.72 ± 2.19	5.72 ± 1.88	0.000	1.000

Data are presented as mean ± standard deviation.

**Table 5 ijerph-17-03263-t005:** The long-term effect of school-based meditation on smartphone addiction, self-control, stress coping strategies, and stress scores.

Characteristics	Categories	Time	Exp	χ^2^(*p*)	Con	χ^2^(*p*)
smartphone addiction	Daily life disturbance	Pretest ^a^	11.96 ± 2.87	25.039 (<0.001)	10.60 ± 3.08	1.463 (0.481)
		Post-test ^b^	10.54 ± 2.03		9.96 ± 2.89	
		Follow-up test ^c^	9.25 ± 2.05		9.92 ± 2.93	
	Directivity to virtual world	Pretest	3.46 ± 1.06	4.246 (0.120)	3.44 ± 0.92	3.447 (0.178)
		Post-test	3.13 ± 1.04		3.36 ± 0.95	
		Follow-up test	2.79 ± 0.88		3.16 ± 1.11	
	Withdrawal	Pretest	7.92 ± 2.96	7.303 (0.026)	8.04 ± 2.48	2.447 (0.294)
		Post-test	6.83 ± 2.32		7.84 ± 2.29	
		Follow-up test	6.08 ± 1.56		7.32 ± 2.23	
	Tolerance	Pretest	10.54 ± 2.21	18.759 (<0.001)	8.64 ± 1.89	0.861 (0.650)
		Post-test	9.13 ± 1.83		8.44 ± 2.10	
		Follow-up test	8.25 ± 2.09		8.40 ± 2.43	
	Total score	Pretest	33.88 ± 7.19	27.482 (<0.001)	39.00 ± 7.13	3.129 (0.209)
		Post-test	29.63 ± 5.72		29.60 ± 7.25	
		Follow-up test	26.38 ± 4.69		28.80 ± 7.66	
Self-control	Instant satisfaction	Pretest	34.42 ± 4.57	33.250 (<0.001)	35.96 ± 5.45	32.804 (<0.001)
		Post-test	37.75 ± 5.28		34.84 ± 6.73	
		Follow-up test	23.13 ± 6.82		24.32 ± 5.58	
	Long-term satisfaction	Pretest	30.88 ± 5.74	11.109 (0.004)	32.08 ± 4.46	1.978 (0.372)
		Post-test	33.67 ± 4.57		32.92 ± 5.29	
		Follow-up test	34.38 ± 4.83		32.24 ± 4.87	
Stress coping strategies	Problem focusing	Pretest	10.25 ± 2.29	15.258 (<0.001)	11.56 ± 2.60	2.422 (0.298)
		Post-test	12.29 ± 3.32		12.12 ± 2.88	
		Follow-up test	12.75 ± 2.69		12.28 ± 3.48	
	Social support navigation	Pretest	12.29 ± 2.16	8.023 (0.018)	12.52 ± 3.31	0.588 (0.745)
		Post-test	14.00 ± 3.56		12.68 ± 3.44	
		Follow-up test	12.88 ± 3.23		13.76 ± 3.80	
	Wishful thinking	Pretest	12.79 ± 1.96	2.868 (0.238)	12.20 ± 2.94	0.156 (0.925)
		Post-test	13.25 ± 2.31		12.00 ± 2.90	
		Follow-up test	12.79 ± 2.28		12.44 ± 3.04	
	Emotion focusing	Pretest	10.38 ± 2.83	1.744 (0.418)	10.24 ± 2.30	12.023 (0.002)
		Post-test	11.67 ± 2.60		10.80 ± 2.36	
		Follow-up test	10.63 ± 2.04		11.80 ± 2.78	
Stress level	Stress level	Pretest	7.79 ± 1.59	32.519 (<0.001)	5.72 ± 2.19	0.030 (0.985)
		Post-test	5.67 ± 2.18		5.72 ± 1.88	
		Follow-up test	4.54 ± 2.38		5.72 ± 2.42	

Data are presented as mean ± standard deviation. ^a^ Pretest scores. ^b^ Post-test scores. ^c^ Follow-up test scores.

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
