# Peer review of "The Effect of Mind Subtraction Meditation Intervention on Smartphone Addiction and the Psychological Wellbeing among Adolescents"

_ijerph, 2020, doi:10.3390/ijerph17093263_

Round 1
Reviewer 1 Report
This work looked at using mind subtraction meditation as a means to reduce smartphone addiction in teens. The statistical treatment of the data was sound. Overall, it was a very interesting study. I would be very curious to see a measure of smartphone usage before and after the study (not just the addition measures) to see if your intervention technique would also be applicable for use to reduce smartphone usage. I have noted some minor issues below:
Minor issues:
- Remove highlighting on line 201
- There are several instances of very short paragraphs that do not transition well to the next (ex. - first 2 paragraphs of discussion). These disrupt the flow of the paper and are a bit distracting.
- Due to lack of borders, tables were hard to read (add borders to rows that mark the end of a measure?)
Author Response
I would like to extend my deep appreciation to you for review.
I thankfully found this round of editing a good chance of improving the paper.
|
Reviewer’s comment |
Response |
|
|
1 |
Remove highlighting on line 201
|
With due respect, I carefully did my best in modifying it. è We deleted highlighting and we changed spacing. |
|
2 |
There are several instances of very short paragraphs that do not transition well to the next (ex. - first 2 paragraphs of discussion). These disrupt the flow of the paper and are a bit distracting.
|
Thank you for your advice. As your advice, we extensively modified this manuscript and we will have professional service to proofread the text before publication. Ex) First 2 paragraphs of discussion This study aimed to verify the effects of Mind Subtraction Meditation on the smartphone addiction, stress level, self-control, and stress coping strategies of adolescents in their second year of high school. The results of this study showed that Mind Subtraction Meditation led to a decrease in smartphone addiction. A prior study that used Mind Subtraction Meditation as an intervention for smartphones in children also showed that the program improved smartphone addiction and mental health by increasing self-understanding and self-review of current stress levels [11]. è Recently, school-based meditation programs could be one of innovative strategies for students with academic, psychosocial, and behavioral demands. This study is one of the few intervention studies that verified the effect of a school-based meditation program on smartphone addiction, stress level, self-control, and stress coping strategies of high school students. The meditation program was offered to the students twice a week, 20 minutes each time during their morning meetings, for a total of 12 weeks. In this study, we found mind subtraction meditation led to a significant decrease of daily life disturbance, withdrawal symptom and tolerance caused by smartphone addiction. Even after one month of completing the program, the total scores of smartphone addiction decreased for experimental group who participate in meditation program. This beneficial effect of the meditation program was also noted in other research studies related to meditation [11, 23]. A prior study showed that a school-based mind subtraction meditation had positive effect on smartphone addiction tendency and addiction associated mental health of children in South Korea [11]. |
|
3 |
Due to lack of borders, tables were hard to read (add borders to rows that mark the end of a measure?) |
è We added a few borders to read easily. |

Reviewer 2 Report
Nowadays, smartphone addiction is a serious social problem that has several adverse consequences for adolescents, especially psychosocial consequences, however, in the introduction section of this paper is not emphasize the seriousness of this problem. That is, the article begins by talking about gambling addiction, but this is not the goal of the paper. In addition, there is also no exhaustive review of the relationship between smartphone addiction and Mind Subtraction Meditation.
Regarding the sample, it was not selected with any random method, and since it belongs to the same center, there is a risk of bias.
Sociodemographic variables were introduced to analyze homogeneity, however, these variables are not mentioned in the methodology section, nor are they used in subsequent analyzes. In other words, sex, religion, etc. are taken into account, but ANOVA analyzes are not performed with the following variables.
The long-term effect was performed, but how was analyzed this long-term effect?, for example, When was the long-term effect performed after finishing the Mind Subtraction Meditation sessions?
Author Response
I would like to extend my deep appreciation to you for review.
I thankfully found this round of editing a good chance of improving the paper.

Reviewer 3 Report
Title: The effect of Mind Subtraction Meditation intervention on smartphone addiction and psychological wellbeing among adolescents
This study explores the mind subtraction mediation intervention impact on reducing smartphone use and improving psychological wellbeing among Korean adolescents. I have some comments for authors to improve the quality.
Major comments:
- the language usage of the manuscript should be extensively modified, please have a native speaker or professional service to proofread the text before publication.
- the rational of this study seems too weak, please add more study rationales into the section of introduction, including relevant literature review, gaps in the literature and significance of this paper. These information would be beneficial to improve the paper considerably.
- as you mentioned in the method section “at the same high school whose classroom was located far from the experimental”, how could you guarantee there was no interaction between experimental and control groups’ participants? Please clarify it.
- in each group, girls exceeded boys significantly greatly, was this resulting in influence on study effectiveness or study results? Please clarify it.
- please define the missing data in line 85? Was it as same as drops of study?
- please report the psychometric values of the scale [13], see line 113.
- see my above comment, please clearly report the psychometric values of all the measures if necessary and wherever.
- line 74, could define the “reading sessions” clearly? what is the difference among it and intervention programs? Please clarify it.
- the current version of discussion is too long, which should be shorten based on main findings. Please revise it.
Minor comments:
- Line 14: the fastest and the most widespread may be too arbitrary, please revised the statements.
- p value, sometimes the p is upper case, while is not. Please unify it.
- the reference should be formed in line with journal’s styles.
- please add a section of study limitation.
Author Response

(The authors gave the same response as above.)

Round 2
Reviewer 2 Report
Every change request has been added to the manuscript.
Reviewer 3 Report
The author has made appropriate modifications to the article as suggested, and I suggest that the research article can be accepted.